# Metabolism in Sync: The Circadian Clock, a Central Hub for Light-Driven Chloroplastic and Mitochondrial Entrainment

**DOI:** 10.3390/plants14162464

**Published:** 2025-08-08

**Authors:** Luis Cervela-Cardona, Marta Francisco, Åsa Strand

**Affiliations:** 1Misión Biológica de Galicia, Consejo Superior de Investigaciones Científicas (CSIC), Apartado 28, 36080 Pontevedra, Spain; mfrancisco@mbg.csic.es; 2Umeå Plant Science Centre, Department of Plant Physiology, Umeå University, 901 87 Umeå, Sweden; asa.strand@umu.se

**Keywords:** circadian clock, metabolic entrainment, chloroplast–mitochondria crosstalk, retrograde signaling, light signaling, plant energy metabolism

## Abstract

Plants align their physiology with daily environmental cycles through the circadian clock, which integrates light and metabolic signals to optimize growth and stress responses. While light entrainment has been extensively studied, emerging evidence highlights the central role of metabolism—particularly from chloroplasts and mitochondria—in tuning circadian rhythms. In this review, we explore the bidirectional relationship between organelle metabolism and the circadian clock, focusing on how metabolic signals such as sugars, ROS, and organic acids function as entrainment cues. We discuss how the clock regulates organelle function at multiple levels, including transcriptional, translational, and post-translational mechanisms, and how organelle-derived signals feedback to modulate core clock components through retrograde pathways. Special attention is given to the integration of chloroplast and mitochondrial signals, emphasizing their synergistic roles in maintaining cellular homeostasis. Drawing on the “three-body problem” analogy, we illustrate the dynamic and reciprocal interactions among light, clock, and metabolism. This perspective underscores the need to reframe the circadian system, not merely as light-driven but also as a central integrator of energy status and environmental cues. Understanding this integrated network is essential to improve plant performance and resilience under fluctuating environmental conditions.

## 1. Introduction

Plants, as sessile organisms, have acquired an ability to fine-tune their growth, development, and metabolism in response to the ever-changing conditions around them. Central to this adaptation is their capacity to perceive and respond to light, a primary source of energy and a critical environmental cue. Light not only drives photosynthesis, the core of plant metabolism and growth, but it also serves as a key signal for the timing of biological processes. Light signaling is intricately linked to the endogenous cell autonomous timekeeper, the circadian clock, which orchestrates a wide array of physiological activities to align the biological rhythms with the day–night cycle [1]. The circadian clock enables plants to anticipate daily changes and prepare for them in advance, optimizing plant survival and fitness. It regulates the expression of numerous genes, gating key processes such as photosynthesis, hormone signaling, and leaf movement to specific times of the day [2,3]. The light perception system and the circadian clock coordinate the adjustment of plant metabolic processes [4]. Simultaneously, metabolic feedback can influence the clock, allowing it to adjust its pace based on internal metabolic states and external environmental changes. This synchronization ensures that energy-intensive activities occur at the most favorable times, leveraging light availability for maximal efficiency. However, the mechanisms by which light signaling and the circadian clock interact to control and adjust metabolism are complex and not fully understood. This is especially critical as plants face the challenges of climate change, with shifting light patterns and temperatures affecting their growth and survival [5]. Exploring how these systems integrate to synchronize metabolic activities with environmental cycles could open avenues for enhancing crop resilience and productivity in an era of global environmental change.

In this review, we propose that metabolic signals from the organelles serve as key entrainers of the circadian clock, which in turn fine-tunes organelle function. Both mechanisms are influenced by light; however, light primarily acts as a regulatory modulator rather than the central mechanism. This highlights the complex interplay whereby organelle metabolism and the circadian clock operate in a dynamic, interconnected system, with light modulating these interactions to optimize the physiological responses of the plant.

## 2. The Plant Circadian Clock

The structure and composition of the plant circadian clock have been extensively studied since the late 1990s (reviewed in [6,7,8,9]). Briefly, the *Arabidopsis thaliana* (Arabidopsis from now on) circadian clock is composed of a complex transcriptional-translational feedback loop (TTFL) that integrates environmental cues to regulate physiological and developmental processes [8,10]. At its core, the clock consists of a morning loop and an evening loop. In the morning, *CIRCADIAN CLOCK ASSOCIATED 1* (*CCA1*) and *LATE ELONGATED HYPOCOTYL* (*LHY*), two MYB transcription factors, repress the expression of evening-phased genes, including *TIMING OF CAB EXPRESSION 1* (*TOC1*, also known as *PSEUDO-RESPONSE REGULATOR 1* (*PRR1*)), *PRR5*, *PRR7*, and *PRR9*. As the day progresses, PRRs sequentially inhibit *CCA1* and *LHY*, allowing the accumulation of the evening complex (EC) components, including *EARLY FLOWERING 3* (*ELF3*), *EARLY FLOWERING 4* (*ELF4*), and *LUX ARRHYTHMO* (*LUX*), which collectively suppress *PRR* expression during the night. In addition, other clock-associated components such as LIGHT-REGULATED WD (LWD) [11], REVEILLE8 (RVE8) [12], and ZEITLUPE (ZTL) [13], among other [14,15,16] proteins, are essential for maintaining the robustness and precision of circadian rhythms (Figure 1). This tightly regulated multilayered feedback network ensures a ~24 h rhythmicity. Additionally, post-translational modifications, such as phosphorylation [17], ubiquitination [18], and metabolic signaling pathways (e.g., sugar and hormone signaling) [19], fine-tune circadian oscillations, enabling the clock to synchronize with environmental fluctuations, particularly light and temperature cycles [20].

### 2.1. Light and Clock: A Bidirectional Relationship

Light entrains the circadian clock through a specialized photoreceptor network, integrating environmental light cues into endogenous rhythms. Arabidopsis photoreceptors belong to the following five families: phytochromes (PHYs), cryptochromes (CRYs), ZTL, phototropins, and UVR8. Except for UVR8, all others associate with chromophores that absorb light and activate signaling pathways. These receptors detect light intensity, quality, direction, and duration (photoperiod), enabling plants to fine-tune their biological rhythms (reviewed in [23,24]).

While photoreceptors enhance entrainment precision, they are not essential for maintaining circadian rhythmicity. Mutants lacking PHYs (PHYA, PHYB) and CRYs (CRY1, CRY2) retain circadian oscillations after light–dark entrainment and under free-running conditions (i.e., in the absence of external time cues such as light or temperature cycles) [25,26,27], indicating photoreceptors are not fundamental components of the core circadian machinery.

In addition, PHYTOCHROME-INTERACTING FACTORS (PIFs), key transducers of phytochrome signaling [28] also mediate sucrose-induced circadian gene expression, highlighting a link between light and metabolic entrainment pathways. Short et al. [29] demonstrated that pifQ (*pif1*, *pif3*, *pif4*, *pif5* quadruple mutant) has a longer circadian period than the wild type under continuous conditions (constant light or constant darkness) when sucrose is present. Specifically, sucrose enhances PIF binding to *CCA1* and *LHY* promoters at subjective dawn, leading to elevated expression of these genes. This induction is delayed in pifQ mutants, highlighting the role of PIFs in sucrose-mediated circadian entrainment, independent of direct light perception [30].

### 2.2. Clock Modulation of Light Signaling and Sensitivity

Beyond photoreceptors acting as transducers of light cues, the circadian clock reciprocally modulates plant sensitivity to environmental light signals, optimizing physiological responses to daily environmental changes [9,31]. The clock enhances early morning photosynthetic efficiency by regulating stomatal opening and leaf orientation [32], and it also fine-tunes photoreceptor expression to anticipate and adapt to daily fluctuation [33,34]. A well-characterized example is the regulation of PHYTOCHROME A (PHYA), which mediates far-red light responses essential for seedling establishment and shade avoidance [35]. PHYA signaling requires nuclear transport facilitated by FAR-RED ELONGATED HYPOCOTYL 1 (FHY1) and FHY1-LIKE (FHL) proteins, transcriptionally regulated by FHY3 (FAR-RED ELONGATED HYPOCOTYL 3) and FAR1 (FAR-RED IMPAIRED RESPONSE 1) transcription factors [36,37]. The core clock components CCA1 and TOC1 directly regulate this process, repressing FHY3 and FAR1 to gate PHYA nuclear accumulation and thus modulating far-red sensitivity in a time-dependent manner [38,39,40]. The loss of CCA1 and TOC1 enhances PHYA signaling, whereas their overexpression suppresses it [41]. Additionally, FHY3 and FAR1 contribute to the regulation of *ELF4*, integrating far-red light perception to the circadian clock evening complex [42].

The functional interplay between light and the circadian clock extends beyond photoreception to encompass the regulation of photoprotective processes. Excess light can exceed the photosynthetic capacity of the plant, resulting in photodamage through the accumulation of reactive oxygen species (ROS) [43,44]. To mitigate this, the clock modulates multiple photoprotective mechanisms, including non-photochemical quenching (NPQ) [45,46], carotenoid biosynthesis [47,48,49], and antioxidant systems, including catalases and peroxiredoxins, which exhibit circadian rhythmicity aligned with diel light cycles [22,50]. This temporal regulation enhances photosynthetic efficiency and protects the photosynthetic apparatus from oxidative stress, underscoring the adaptive significance of circadian control in dynamic light environments.

Taken together, these findings reveal a complex regulatory network in which the circadian clock not only receives light input but also modulates light perception, aligning both photoreceptor activity and photoprotective responses with circadian timing. While light is undeniably a primary entraining signal and has been extensively studied in this context, sucrose metabolism also emerges as a potent circadian cue. Nonetheless, the role of metabolism in circadian entrainment has received comparatively less attention, partly due to the intrinsic coupling of metabolic activity with light availability in plant systems. The following sections will examine how metabolic signals derived from chloroplasts and mitochondria contribute to circadian regulation and how, in turn, the clock coordinates organelle function to optimize physiological responses.

## 3. Clock and Metabolism: Ticking Organelles

Light is a major entrainer of the circadian clock; however, it is not the sole factor driving rhythmicity. The clock can maintain stable oscillations even in the absence of photoreceptors (see Section 2), suggesting that additional cues contribute to its regulation. Metabolic signals, particularly those derived from sugar metabolism, play a crucial role in this process. In total darkness, the application of exogenous sucrose can sustain rhythmic expression of core clock components such as *TOC1* and *GI*, and their phase is set by the time of sucrose addition [51,52].

Notably, excised roots retain circadian rhythms for up to four days when supplied with sucrose [53]. Consistently, sucrose accumulation in roots follows a circadian oscillation, and root *PRR7*, *PRR5*, and *PRR9* expression responds dynamically to sucrose levels [54]. In contrast, whole-plant analyses revealed that while shoot *PRR7* is regulated by sucrose, *CCA1*, *PRR5*, and *PRR9* do not exhibit similar sucrose responsiveness [55]. This spatial divergence suggests that sucrose entrains the circadian clock in a tissue-specific manner, independent of direct light perception. Roots, which are not exposed to light, rely on shoot-derived sucrose as a systemic signal—supporting its role as a non-photic entrainment cue [30].

Furthermore, the clock component *PPR7* is repressed by sugars at night. This process is mediated by the BASIC LEUCIN ZIPPER 63 (bZIP63), a transcription factor that binds directly to *PPR7* promoter, upregulating its expression under low energy conditions (e.g., low sugar, low light) [56]. The activity of bZIP63 is regulated by SNF1 kinase homolog 10 (KIN10), the catalytic subunit responsible for most of the SNF1-RELATED PROTEIN KINASE 1 (SnRK1) activity. SnRK1 is the plant ortholog of the mammalian cellular energy sensor AMP-activated protein kinase (AMPK) which regulates the circadian clock in response to energy status [57,58]. The sugar-dependent regulation of *PRR7* by bZIP63 allows the clock to dynamically adjust to cellular energy status, reinforcing the critical role of metabolism in circadian entrainment. Additionally, sugars modulate the circadian oscillator post-translationally by stabilizing GIGANTEA (GI) in a ZEITLUPE (ZTL)-dependent manner. Similarly to blue light, sucrose promotes the GI–ZTL interaction during the evening, enhancing the degradation of clock components such as TOC1 and PRR5 via the 26S proteasome [59] (Figure 1). This stabilization mechanism fine-tunes the amplitude of evening clock output and illustrates how metabolic and light signals converge on shared post-translational regulators to coordinate circadian timing.

The circadian clock also regulates flowering and growth, which are influenced by photoperiodic cues such as light duration. These developmental transitions are controlled through the CONSTANS (CO) and FLOWERING LOCUS T (FT) pathway [60,61,62]. While interconnected, the timing of growth and flowering varies across the year. For instance, plants often experience rapid vegetative growth under short days but flower more quickly under long days [63]. This differential timing in flowering and growth is due to the following two distinct mechanisms for photoperiodic measurement: the CO/FT regulon and the metabolic day length measurement (MDLM) system. The CO/FT regulon relies on photoreceptors to measure the absolute photoperiod (total light hours) [64]. In contrast, the MDLM system allows plants to measure effective photosynthetic duration independently of direct light perception (photosynthetic period). This is achieved through chloroplast metabolic activity, where sucrose and starch production regulate gene expression under both short- and long-day conditions in a CO-independent manner [65,66]. This agrees with previous studies showing the relevance of photoassimilates in the entrainment of the circadian clock [67].

The two main organelles for energy metabolism in plants are mitochondria and chloroplasts. These organelles give plants the ability to generate carbon skeletons and consume them for growth and development. The circadian clock exerts tight control over both organelles by regulating specific target proteins and processes (Figure 2). It is crucial to bear in mind that chloroplasts and mitochondria are closely interconnected as they work together through several metabolic pathways [68,69,70]. The intricate interplay between the circadian clock and organelles ensures optimal adaptation to environmental changes. The entangled relationships involve multiple levels of regulation and are bidirectional, with signals from the nucleus to the chloroplasts and mitochondria (anterograde signaling), and from the organelles back to the nucleus (retrograde signaling) [71,72,73] and between organelles (crosstalk) [74]. The following sections will explore how the circadian clock governs organelle function (Section 3.1 and Section 3.2) and, conversely, how organelle-generated metabolic signals serve as entrainment cues that reinforce circadian rhythmicity (Section 3.3). This bidirectional regulation highlights the intricate metabolic feedback loops that synchronize cellular processes with environmental cycles.

### 3.1. Circadian Control of Chloroplast

The circadian clock affects chloroplast performance, development, and biogenesis. Early transcriptomic analyses revealed that many genes involved in light-harvesting reactions of photosynthesis are under circadian control. For example, transcript levels of the *LIGHT-HARVESTING ANTENNA COMPLEX* (*LHC*) rise before dawn and peak in the morning [75]. This circadian regulation is modulated by the direct binding of circadian proteins such CCA1 and LHY to the promoter region of *LHC* genes such *LHCB1.3* and *LHCB1.1* [76,77,78]. Conversely, chloroplast can also affect clock pace. Up to 70% of chloroplast-encoded genes in Arabidopsis exhibit circadian rhythmic expression [79]. Additionally, it is well established that the circadian clock controls many aspects linked to the photosynthetic process beyond the light reactions of photosynthesis [80,81] such as net CO_2_ uptake [82,83], stomatal opening–closing [32,84], and chloroplast movement [85].

An early example of circadian control over chloroplast function is the CHLOROPLAST RNA BINDING (CRB) protein, which is regulated by the circadian clock. Mutants lacking *CRB* exhibit severe impairments in chloroplast photosynthetic performance [86]. Moreover, these mutants display a delayed photoperiod and increased amplitude of *CCA1*, *LHY*, and other clock output genes, potentially due to alterations in sugar-mediated signaling from the chloroplast [86,87]. This supports the role of chloroplasts not only as passive recipients of circadian regulation but also as environmental sensors that transmit signals to the nucleus via retrograde signaling [88,89]. Numerous studies have documented chloroplast-to-clock retrograde signaling, revealing multiple levels of regulation (see Section 3.3.1).

Another example is the control of chlorophyll biosynthesis by the circadian clock. The first step of the chlorophyll branch is the ATP-dependent insertion of the Mg^2+^ ion into protoporphyrin IX, a reaction that is catalyzed by magnesium chelatase (MgCh) [90]. The CHLH (also known as GENOMES UNCOUPLED 5 [GUN5]) is one of the three subunits of the MgCh [91]. Its expression is under circadian regulation, with TOC1 directly binding to the *GUN5* promoter at night, repressing its transcription (Figure 2a). This TOC1-mediated inhibition ensures rhythmic control of *GUN5* expression, linking the circadian clock to chlorophyll biosynthesis [92].

Besides the direct regulation by clock proteins to chloroplast machinery, other levels of regulation have been described. In plants, SIGMA FACTORS (SIGs) facilitate promoter recognition and transcription initiation by the PLASTID-ENCODED RNA POLYMERASE (PEP), regulating chloroplast biogenesis and steady-state photosynthesis [93]. The Arabidopsis nuclear genome encodes six SIGs (SIG1 to SIG6) all of which are subject to circadian regulation. Among them, *SIG1*, *SIG3*, *SIG4*, and *SIG5* exhibit transcript oscillations, transmitting circadian signals to their chloroplast target genes, such as *psaA*, *psbN*, *ndhF*, and *psbD*, [94,95,96] (Figure 2a). Notably, *SIG5* plays a dual role as follows: it integrates light signals to regulate chloroplast-encoded gene expression based on light intensity and quality [97], and it contributes to cold-response modulation, ensuring photosynthetic efficiency under low-temperature conditions through a mechanism involving ELONGATED HYPOCOTYL5 (*HY5*) and HY5 HOMOLOG (*HYH*) [98].

Since the first transcriptomic studies where rhythmical patterns in gene expression were found, more intricate layers of the clock regulation of chloroplast function have emerged. A key feature of the chloroplast is the capacity to perform photosynthesis. Hence, the circadian regulation of both photosynthesis-associated plastid genes (PhAPGs) and photosynthesis-associated nuclear genes (PhANGs) plays a crucial role [87,99,100]. Several transcription factors are essential to drive the expression of *PhANGs*, for example, GOLDEN2-LIKE (GLK) TFs are indispensable for chloroplast biogenesis. In Arabidopsis the *GLK* genes exist as a pair named *GLK1* and *GLK2* [101]. Arabidopsis *glk1 glk2* double mutants are pale green, and mesophyll cells contain small chloroplasts with sparse thylakoid membranes that fail to form proper grana [102,103]. *GLK2* is directly activated by CCA1, showing a mechanism by which the clock can modulate *PhANGs* and thereby chloroplast function [104] (Figure 2a). In addition to this direct link between the clock and regulation of chloroplast machinery, a novel regulator of chloroplast development, BRZ-INSENSITIVE-PALE GREEN 4 (BPG4), has proven to be under circadian control. BPG4 suppresses GLK2 activity by inhibiting GLK2 DNA-binding activity, reducing the expression of *PhANGs* and leading to decreased chlorophyll biosynthesis and LHC antenna size. The circadian clock’s regulation of BPG4 aligns the expression of *PhANGs* with the day–night cycle. This coordination ensures that chloroplast development is optimized for the photosynthetic needs of the plant at different times of the day, potentially maximizing efficiency and avoiding unnecessary energy expenditure [105].

In *Ostreococcus tauri*, the smallest known photosynthetic eukaryote, which contains a single chloroplast and a single mitochondrion, the relationship between circadian regulation of chloroplastic and mitochondrial transcript and protein abundance is only partial, similar to what is observed in Arabidopsis [106]. Still, in *O. tauri*, a few chloroplast-encoded proteins associated with photosynthesis, transcription, and translation exhibit circadian regulation [107], supporting the notion that the circadian regulation of chloroplast machinery is conserved across the green lineage.

The impact of the circadian clock on the regulation of chloroplast function is well-established and multileveled, demonstrating how this biological timing system precisely aligns chloroplast activities with the day–night cycle. This regulation ensures that photosynthetic activity, chloroplast development, and biogenesis are synchronized with environmental rhythms, optimizing plant energy production. However, while the circadian regulation of chloroplasts is well established, the role of the clock in regulating the plant mitochondria—the cellular powerhouses responsible for respiration and energy production—remains far less understood.

### 3.2. Circadian Control of Mitochondria

In mammals, the connections between the circadian clock and mitochondrial metabolism have been extensively studied and described as crucial circadian entrainers (reviewed in [108,109]). Fully functional mitochondria are essential for maintaining rhythmic circadian clocks in mammals [110]. Moreover, the circadian clock exerts a tight control of mitochondrial function at many levels. For instance, a wide range of metabolites associated with mitochondrial functions such as ATP levels [111], ROS [112], nicotinamide adenine dinucleotide (NAD) [113,114], and tricarboxylic acid cycle (TCA) intermediates [115] exhibit circadian rhythmicity in animal models. Furthermore, mitochondrial dynamics, essential for proper functionality, are also clock-dependent [112]. Interestingly, disruptions in metabolic homeostasis due to high-fat and -sugar diets in mammals are associated with disturbances in circadian rhythms and conversely, misregulation of the circadian clocks has a critical impact on cellular metabolism, highlighting the reciprocal relationship between the metabolic status of the organism and the circadian clocks [116]. Similarly, the disruption of proper mitochondrial function severely impacts plant performance. For instance, depletion of certain mitochondrial-related genes causes a plethora of phenotypes ranging from embryo lethality to an array of mild to severe growth phenotypes [117,118,119,120,121,122,123,124,125], stressing the importance of this organelle for plant biology. However, in plants, studies on the circadian regulation of mitochondrial function are limited.

Remarkably, c.a. 65% of the genes that encode for mitochondrial proteins in the nucleus display diel oscillation in transcript abundance, many of which peak at night [126]. In addition, the circadian clock regulates the ribosomal loading of mRNAs, thereby influencing their translational state. Notably, mRNAs encoding mitochondrial respiration components exhibit peak ribosome association during the night, leading to enhanced protein synthesis at this time [127].

Similarly to *O. tauri*, where only two mitochondrial proteins exhibit rhythmic accumulation under constant light [107], and in contrast to chloroplasts, only a few nuclear-encoded mitochondrial-related proteins show circadian accumulation patterns in Arabidopsis [128,129,130]. This rhythmicity in protein abundance is partially modulated by the circadian-regulated TCP transcription factor family (TEOSINTE BRANCHED 1 (TB1) in maize, CYCLOIDEA (CYC) in snapdragon, and PROLIFERATING CELL FACTOR 1 and 2 (PCF1/2) in rice) [131]. An alternative mechanism for regulating diel mitochondrial activity has been proposed, suggesting that dynamic protein–protein interactions within mitochondrial complexes change between light and dark periods, offering a new avenue for mitochondrial regulation [130]. Comparative analysis of RNA-seq and protein mass spectrometry in a subset of clock mutants has shown misregulation of a group of mitochondrial-related genes and proteins specific to each of the core clock component, although there was little correlation between transcript and corresponding protein accumulation, suggesting another layer of regulation based on posttranslational modifications rather than direct protein turnover [132].

In addition to mitochondrial-related transcripts and proteins showing circadian accumulation, byproducts of the mitochondrial metabolism also display a circadian signature, so relevant that it has been proposed that changes in metabolites drive gene expression and not the other way around [133]. Metabolites produced by the TCA display robust rhythmic accumulation matching the day–night cycle [134,135,136] (Figure 2b). This diel accumulation of TCA intermediates is disrupted when a functional circadian clock is missing, as in the *prr579* triple mutant and *cca1*/*lhy* double mutant [137,138]. One example of this circadian TCA regulation is the rhythmic accumulation of fumarate which is regulated by the direct binding of TOC1 to the promoter region of the *FUMARASE 2*, a key enzyme in TCA cycle, giving the plant diel control of the carbon and nitrogen pools [126].

Furthermore, mitochondria are the site of amino acid catabolism [139,140,141,142]. During seed germination and development, dark-induced starvation and water deprivation conditions, catabolism of amino acids such as proline, lysine, and branched chain amino acids (BCAA) can support ATP generation through providing electrons to the alternative electron-transfer flavoprotein/electron-transfer flavoprotein:ubiquinone oxidoreductase (ETF/ETFQO) complex pathway [143,144,145]. Several genes encoding BCAA catabolic enzymes exhibit strong diel and circadian expression profiles showing a transcript enrichment during the night [146], suggesting they are regulated by light and the circadian clock. This rhythmical accumulation is in agreement with the observations of the circadian accumulation of free BCAA that peaks at the end of the day and is dampened at the end of the night [133,147], indicating that this process is under circadian clock influence.

### 3.3. The Organelle Entrainment of the Clock

The circadian clock exerts tight control over the organelles in the cell; however, these organelles produce byproducts (e.g., sugars, organic acids, ROS, and ATP) that, in turn, influence circadian function. The following sections will explore how chloroplast metabolism contributes to circadian entrainment (Section 3.3.1), how mitochondria generate rhythmic metabolic cues (Section 3.3.2), and how the coordination between the two organelles fine-tunes clock regulation (Section 3.3.3).

#### 3.3.1. Chloroplast Entrainment

Sugars derived from the photosynthetic process in chloroplasts entrain the circadian clock differently from light (Section 3). This sugar-mediated entrainment is a gradual process that occurs when sugar levels, produced within the chloroplasts, reach a metabolic dawn at which they are sufficiently high to influence the circadian clock [55]. The effects of sugars on the circadian clock are complex, and we are just starting to elucidate the mechanisms by which sugars entrain the clock. For instance, Haydon et al. [55] demonstrated that the application of exogenous sugars shortened the period of the core clock gene *CCA1*. High sugar levels in the evening, originating from the chloroplast during active photosynthesis, can delay the clock phase by repressing evening complex genes such as *ELF3* and *LUX*, thereby extending the active phase of photosynthesis and aligning the plant’s metabolic status with energy availability [55,56]. Additionally, the SnRK1 pathway, interacting with trehalose-6-phosphate (T6P) and the TARGET OF RAPAMYCIN (TOR) signaling pathway, integrate these sugar signals from the chloroplasts to further modulate the circadian clock. SnRK1 adjusts circadian responses by linking sugar availability with cellular energy demands [148,149], while TOR acts as a critical mediator that integrates environmental and metabolic cues to regulate the clock’s speed and synchronization with plant growth cycles [150,151].

Building on this understanding, Boix et al. [152] demonstrated that the 40S ribosomal protein S6 kinase (S6K1), a component of the TOR pathway, is regulated by the circadian clock and plays a significant role in synchronizing growth with metabolic status. S6K1 levels oscillate under different photoperiods, peaking during the light period and decreasing at dusk, particularly under short day conditions. This pattern was disrupted in mutants lacking the circadian F-box protein ZTL, indicating that S6K1 serves as a key integrator of metabolic and circadian signals, thereby coordinating growth with resource availability. The study also showed that the oscillations in S6K1 are linked to the regulation of carbon metabolism, including starch, sucrose, and glucose levels, which are crucial for growth [152]. This finding underscores the role of the TOR pathway in modulating the plant’s circadian clock in response to metabolic signals, particularly from sugars. This complex network ensures that circadian rhythms are not only responsive to direct light signaling but are also adjusted in response to light-driven products from the chloroplasts, where sugars are synthesized and serve as key regulatory signals. These metabolic cues influence circadian regulation through mechanisms involving multiple clock components (e.g., PRR7, GI) and energy status signaling pathways (e.g., SnRK1, TOR) [148].

Furthermore, the sugar transport system is also clock-regulated, highlighting the intricate relationship between metabolic activity and circadian rhythms. At night, starch accumulated during the day is primarily degraded via a hydrolytic pathway, producing maltose and a lesser amount of glucose in chloroplasts. These sugars are then exported to the cytosol, where they are converted into sucrose and utilized in respiration [153]. Interestingly, maltose levels oscillate in a circadian manner, even under continuous light, showing a sharp increase after the onset of darkness [154] (Figure 2a). Critical genes encoding chloroplast transporters, such as the *TRIOSE PHOSPHATE/PHOSPHATE TRANSLOCATOR* (*TPT*) and the *MALTOSE EXCESS1* (*MEX1*), exhibit rhythmic expression patterns controlled by the circadian clock. These transporters facilitate the movement of essential metabolites between the chloroplast and cytosol, synchronizing metabolic fluxes with the circadian clock [155].

Nevertheless, the role of chloroplasts in clock entrainment extends beyond the production and export of photoassimilates, as crucial ions such as Fe^2+^, Mg^2+^, and Ca^2+^ contribute to circadian entrainment via retrograde signaling. Fe^2+^ and Mg^2+^ deficiencies prolong the clock period [156,157,158,159], while oscillating Mg^2+^ levels support photosynthetic efficiency [160,161]. Ca^2+^ exhibits intrinsic circadian oscillations driven by cADPR and integrates into the central oscillator via *TOC1*–*CML24* [162,163,164]. The circadian clock also regulates chloroplastic Ca^2+^ fluxes, establishing a bidirectional loop (Figure 2a). These Ca^2+^ rhythms persist in darkness when sucrose is available, indicating light-independent, clock-driven ionic entrainment [165,166].

In *Arabidopsis*, the production of the key defense phytohormone salicylic acid (SA) is initiated in the chloroplast and is under circadian control [167,168], with clock-regulated expression of biosynthetic and transport genes (e.g., ICS1, EDS5, EDS1) [169,170,171]. SA reinforces clock–immune coordination by modulating defense gene expression and feeding back to regulate core clock components such as CCA1 and TOC1, thereby integrating immunity with circadian timing [172,173]

The accumulation of NADP^+^ (nicotinamide adenine dinucleotide phosphate) and its reduced form, NADPH, essential cofactors for maintaining redox homeostasis and photosynthesis in chloroplasts, also follows circadian oscillation. SA-induced ROS production affects the circadian fluctuation of the NADP^+^/NADPH ratio, ultimately modulating *TOC1* expression and shifting the clock’s phase and period [174]. Similarly to sugar-mediated circadian regulation, which enables rhythms to persist even in DD conditions, SA-triggered redox changes reinforce circadian rhythmicity in the absence of external light cues [174].

Additionally, the chloroplast redox cycle, mediated by NADPH-dependent thioredoxin reductase type C (NTRC), has been recently shown to couple chloroplast metabolic oscillations with nuclear circadian components [175]. NTRC regulates intracellular ROS and sucrose levels [176], which in turn modulate the amplitude and period of key circadian genes such as *CCA1* and *GI*. Mutants lacking *NTRC* exhibit disrupted nuclear clock oscillations, highlighting a direct metabolic influence on circadian regulation. Furthermore, the nuclear oscillator TOC1 reciprocally regulates *NTRC* expression, suggesting a bidirectional feedback loop between the nuclear and chloroplastic rhythms [175] (Figure 2a). This further emphasizes the role of chloroplast metabolic signals in maintaining circadian rhythms beyond photic inputs.

#### 3.3.2. Mitochondrial Entrainment

As described in Section 3.2, abundant evidence demonstrates how the circadian clock regulates mitochondrial function. It is equally true that mitochondrial activity feeds back to the circadian clock, creating a bidirectional regulatory loop. In mammals, changes in mitochondrial metabolism serve as retrograde signals to the circadian clock. Disruptions in oxidative phosphorylation and mitochondrial DNA depletion can alter circadian gene expression [110]. The mitochondrial NAD+ pool oscillates diurnally, influencing the molecular clock by modulating SIRTUIN 1 and 3 (SIRT1 and SIRT3) activities, which affect core clock proteins and histone acetylation [177,178]. Additionally, 5′-ADENOSINE MONOPHOSPHATE-ACTIVATED PROTEIN KINASE (AMPK) activation links nutrient signals to the circadian rhythm by regulating Cryptochrome 1 (Cry1), and Dynamin-related protein 1 (Drp1). Drp1-mediated mitochondrial fission impacts ATP production and circadian period length, further integrating mitochondrial function with circadian regulation [179].

In contrast to the extensive body of work on chloroplast–clock interactions (see Section 3.3.1) our understanding of how mitochondria feed back into the circadian clock remains limited. Nonetheless, emerging evidence underscores the pivotal role of mitochondrial function in proper clock regulation. Pharmacological disruption of mitochondrial activity with inhibitors such as carbonyl cyanide m-chlorophenyl hydrazine (CCCP), affect the circadian rhythm [180]. These pharmacological treatments likely alter mitochondrial redox status, ATP production, and ROS homeostasis, all of which can transmit signals to the nucleus and influence the timing components of the central oscillator’s [151].

Recent work has shown that mitochondrial retrograde signaling plays a crucial role in circadian regulation through the transcription factor NAC DOMAIN-CONTAINING PROTEIN 17 (ANAC017). Under mitochondrial stress, ANAC017 is activated and relocates to the nucleus, where it modulates the expression of key nuclear-encoded mitochondrial proteins [181]. However, rather than being a purely stress-induced mechanism, ANAC017 activity is diurnally regulated, with increased promoter binding during the light period compared to the dark (Figure 2b). Overlapping regulatory sites have been identified for *ANAC017* and core circadian clock components (such as *CCA1/LHY* or *PRRs*) in the promoters of mitochondrial stress-related genes, including *ALTERNATIVE OXIDASE 1* (*AOX1a)* and *ANAC013*. This points to a bidirectional regulatory relationship where, in addition to the influence of the clock on mitochondrial function, a mitochondrial signal helps fine-tune circadian rhythms [182]. These insights reinforce the idea that mitochondria are not just passive recipients of circadian control but actively contribute to clock entrainment, likely through metabolic cues and retrograde signaling mechanisms.

#### 3.3.3. Synergic Entrainment: The Intracellular Dynamic Duo

As autotrophic organisms, plants depend on the intertwined metabolic processes of chloroplasts and mitochondria. Although it is evident that the circadian clock regulates specific pathways within each organelle, understanding the circadian impact on their interplay is fundamental. This inter-organelle dialogue is crucial for maintaining metabolic homeostasis throughout the day [19]. We must also consider how the clock could control their interaction and how this would influence overall plant metabolism and adaptation to environmental cycles. The photorespiratory pathway is a well-known example of multi-organelle engagement, requiring coordination between chloroplasts, peroxisomes, and mitochondria to efficiently recycle 2-phosphoglycolate (2-PG) and prevent its inhibitory effects on metabolism (Figure 2c). This costly process occurs when RIBULOSE-1,5-BISPHOSPHATE CARBOXYLASE/OXYGENASE (RUBISCO) fixes oxygen instead of carbon dioxide, necessitating the recovery of carbon through photorespiration [183,184]. Since early studies, it has been known that several key enzymes in the photorespiratory cycle are under circadian control, ensuring their activity aligns with peak photosynthetic efficiency. In the chloroplast, the RUBISCO SMALL SUBUNIT (RBCS) and RUBISCO ACTIVASE (RCA) show rhythmic expression, regulating carbon fixation capacity [185]. In the peroxisome, CATALASE 2 (CAT2), which detoxifies H_2_O_2_ generated during glycolate oxidation, is also clock-controlled [186,187]. In the mitochondria, SERINE HYDROXYMETHYLTRANSFERASE (SHM), an essential enzyme for glycine-to-serine conversion in the photorespiratory cycle, exhibits circadian oscillations in phase with other photorespiratory genes [188]. These findings highlight the circadian clock’s critical role in synchronizing cellular machinery to maintain carbon and nitrogen balance throughout the day.

The SAL1-PAP-XRN4 retrograde signaling pathway exemplifies the intricate communication between chloroplasts, mitochondria, and the nucleus in the control over the plant circadian clock (Figure 2c). Under normal conditions, SAL1 (a phosphatase located in both chloroplasts and mitochondria) maintains low levels of 3′-phosphoadenosine 5′-phosphate (PAP) by converting it to adenosine monophosphate (AMP) [189,190]. However, during stress conditions, such as drought or high light intensity, ROS accumulation within these organelles inhibits SAL1 activity, leading to PAP accumulation. Elevated PAP levels inhibit 5′-3′ EXORIBONUCLEASES (XRNs), including cytosolic XRN4. This inhibition results in the stabilization of specific mRNAs, limiting the degradation of transcripts of clock genes such as *PRR7* and *LIGHT-REGULATED WD1 (LWD1)*, a key clock-associated regulator involved in transcriptional activation of circadian genes [191,192,193]. The addition of exogenous PAP, as well as mutations in either the *SAL1* or *XRN4* genes, has been shown to lengthen the circadian period [194,195], emphasizing the role of the PAP-mediated pathway in synchronizing internal biological clocks with environmental stress signals through post-transcriptional regulatory mechanisms.

Chloroplasts and mitochondria are major sites of ROS production, acting both as sources of oxidative stress and as key signaling hubs [196,197]. The rhythmic control of ROS production and scavenging is a well-documented phenomenon, establishing temporal coordination between chloroplast and mitochondrial metabolism. This regulation prevents oxidative stress and ensures that redox signals contribute to cellular homeostasis rather than inducing damage [198]. In chloroplasts, ROS production peaks during the day due to high photosynthetic activity, particularly through superoxide (O_2_^−^) generation at Photosystem I (PSI) and singlet oxygen (^1^O_2_) formation at Photosystem II (PSII) [22,199] (Figure 1). To prevent oxidative damage, the circadian clock controls the expression and activity of key ROS-scavenging enzymes, including ascorbate peroxidase (APX) and glutathione peroxidase (GPX), which are rhythmically expressed to match ROS production with detoxification capacity [198]. At night, when photosynthesis stops, chloroplast-generated ROS levels decline, the mitochondrial electron transport chain (ETC) becomes the dominant source of ROS. Mitochondria produce ROS primarily at Complex I and Complex III, generating O_2_^−^ that is quickly converted to H_2_O_2_ [200]. Moreover, the circadian clock regulates mitochondrial antioxidant defense mechanisms, modulating the expression of AOX1a, which mitigates ROS accumulation by providing an alternative respiratory pathway that prevents excessive electron leakage (Section 3.3.2). Additionally, the clock fine-tunes the activity of PRXs [201] and thioredoxin [202] systems, which maintain redox balance in both organelles during the day–night cycle.

Noteworthily, Román et al. [203] demonstrated that sugars can modulate the circadian clock independently of light through ROS, particularly O_2_^−^. By comparing the transcriptomic response to sucrose in darkness and in light conditions where photosynthesis was inhibited, they identified redox and ROS-related processes as key transcriptional responses to sugar availability, independent of light-driven reactions. Their study revealed that sucrose-induced O_2_^−^ accumulation functions as a signaling molecule, altering both gene expression and growth. Specifically, O_2_^−^ signal promotes the transcription of circadian oscillator genes, including *TOC1*, in the evening, unveiling a novel metabolic feedback mechanism where ROS serves as an input to the circadian clock. Given that sucrose enhances mitochondrial respiration [204,205], which is a major source of O_2_^−^ through electron leakage at Complex I and III of the mitochondrial electron transport chain, mitochondria are the most likely origin of this signal. This aligns with recent findings about mitochondria-generated ROS regulating circadian rhythms and metabolic adjustments [182]. These findings underscore the role of ROS as metabolic signals that regulate circadian rhythms in Arabidopsis, independently of photoperiodic cues. This dynamic regulation reinforces the circadian clock as a master integrator of ROS homeostasis, balancing energy production and cellular protection across the diel cycle

Despite the extensive discussion in Section 3 on the coordination between organelles and the circadian clock, the precise mechanisms by which chloroplast- and mitochondria-derived signals are integrated into nuclear gene expression remain largely unknown [206]. One potential regulator in this process is CDKE1 (CDK8), a cyclin-dependent kinase associated with the Mediator complex, which has been implicated in the transcriptional regulation of both chloroplast (*LHCB2.4*) and mitochondrial (*AOX1a*) genes in response to perturbations in electron transport chain (ETC) and oxidative stress [207] (Figure 2c). This regulation occurs through its interaction with KIN10, positioning CDKE1 as a key relay between stress-induced transcription factors and RNA polymerase II, directly influencing stress-responsive transcriptional programs [208]. Interestingly, the sugar-dependent regulation of PRR7 by bZIP63 (see Section 2) is hypothesized to involve not only KIN10 but also additional kinases, suggesting a broader metabolic signaling network that fine-tunes circadian period length [56]. Additionally, SENSITIVE TO FREEZING 6 (SFR6/MED16), another subunit of the Mediator complex, has been shown to mediate sucrose-induced circadian period adjustments, though the precise mechanism remains unknown [209,210]. Notably, CDKE1 also modulates core circadian clock components, such as *LHY*, *CCA1*, and *RVE* genes, suggesting a role in linking circadian transcriptional regulation with metabolic and stress responses [211]. This highlights the Mediator complex as a central hub that integrates metabolic and environmental signals to fine-tune circadian rhythmicity.

## 4. Light, Clock, and Metabolism Integration: The Three-Body Problem

In astronomy, the three-body problem illustrates the challenge of predicting the motion of three celestial bodies influenced by their mutual gravity, often leading to unpredictable, chaotic dynamics (reviewed in [212]). A parallel can be drawn in plant biology regarding the interactions between light perception, the circadian clock, and metabolism. These three components operate in a delicate balance, analogous to an astronomical system where a low-mass body (e.g., an asteroid) is influenced by, but not essential to, the primary system’s dynamics. Similarly, light entrains and fine-tunes circadian and metabolic rhythms but is not strictly necessary for their persistence. This analogy accentuates the complexity of plant systems, where photoreceptors, despite not being core components of the circadian clock, play a crucial role in aligning internal rhythms with external cues (Figure 3).

As discussed in this review, extensive research has explored the interactions between light perception, the circadian clock, and metabolism—three fundamental components essential for plant growth and development. Chloroplasts and mitochondria, as central hubs of energy metabolism, are tightly integrated into this system. Their functions are circadian-regulated, ensuring that photosynthesis, respiration, and metabolic fluxes align with environmental cycles. This synchronization optimizes resource allocation, growth, and stress resilience, allowing plants to anticipate and adapt to fluctuations in light availability and metabolic demands. The circadian clock serves as a central hub, integrating environmental and energetic signals to coordinate chloroplast and mitochondrial metabolism with transcriptional and physiological responses.

The three-body problem analogy also highlights the bidirectional relationship between the circadian clock and organelle metabolism. The clock orchestrates chloroplast and mitochondrial rhythms, ensuring efficient energy production and utilization, while metabolic feedback from these organelles modulates circadian timing. Although light influences this interplay, organelle-derived signals such as sugars and ROS persist in its absence, underscoring the role of metabolism as a primary entrainer. Understanding this reciprocal relationship provides insight into how plants sustain biological rhythms in dynamic environments. However, metabolism may also function autonomously, orchestrating its rhythms independent of the clock, while still fine-tuning circadian timing through chloroplast- and mitochondria-derived feedback mechanisms.

Future research should focus on unraveling how these signaling pathways converge and identifying key regulatory nodes that integrate metabolic and circadian networks. Understanding these connections is essential for determining how plants synchronize internal rhythms with fluctuating environmental and metabolic conditions, ultimately optimizing growth and stress resilience.

## 5. From Circadian Control to Agricultural Innovation

Building upon the central role of circadian rhythms in coordinating chloroplast and mitochondrial metabolism, recent advances highlight the translational potential of circadian regulation for crop improvement. Circadian rhythms orchestrate photosynthesis, nutrient uptake, and stress responses, enabling plants to anticipate and adapt to environmental changes [213]. Disruptions in these rhythms compromise physiological efficiency and resilience, emphasizing the importance of circadian regulation as a target in breeding and agronomy [214]. Domestication and selection have shaped circadian traits to match regional environments. In tomato, for example, domestication selected mutations in *EID1* and *LNK2* that delay circadian phase and extend the internal day, enhancing performance under high-latitude summer conditions [215]. Similar patterns are observed in soybean and Brassica cultivars, where extended circadian periods correlate with increased biomass and latitude adaptation [216,217]. In parallel, photoperiodic flowering regulation via *ELF3*, *LUX*, *PRR3/7*, and *GI* has been widely targeted to modulate heading date and yield stability [218,219,220,221,222]. These insights have catalyzed the development of chronoculture, an approach that aligns agricultural practices with the plant circadian clock [5]. Notably, crops engineered with clock mutations such as *CCA1* or light-responsive loci like *CAB13* show enhanced yield under continuous or artificial light [223], demonstrating the potential of synchronizing internal rhythms with controlled environments, including vertical farming. Furthermore, precision agriculture strategies incorporating circadian timing—such as timed application of water, fertilizers, or stress signals—could improve uptake efficiency and minimize input waste [224]. Circadian-regulated promoters [225] and master regulators integrating chloroplast- and mitochondrial-derived metabolic feedback [70] represent promising tools for enhancing stress resilience in fluctuating climates.

Finally, circadian heterosis—non-additive interactions in clock gene expression in hybrids—offers a promising strategy to enhance photosynthesis and starch metabolism through optimized temporal regulation [63]. Understanding how the circadian clock acts as a regulatory hub across genotypes may unlock new potentials in hybrid breeding.

By integrating knowledge from plant physiology, genomics, and synthetic biology, future research can move toward engineering crops with enhanced energy efficiency, stress resilience, and adaptive capacity. The discovery of a central regulatory hub that integrates light, circadian, and metabolic pathways could revolutionize agricultural biotechnology, opening new possibilities for the way for sustainable crop production in an era of global climate change.

## Figures and Tables

**Figure 1 plants-14-02464-f001:**
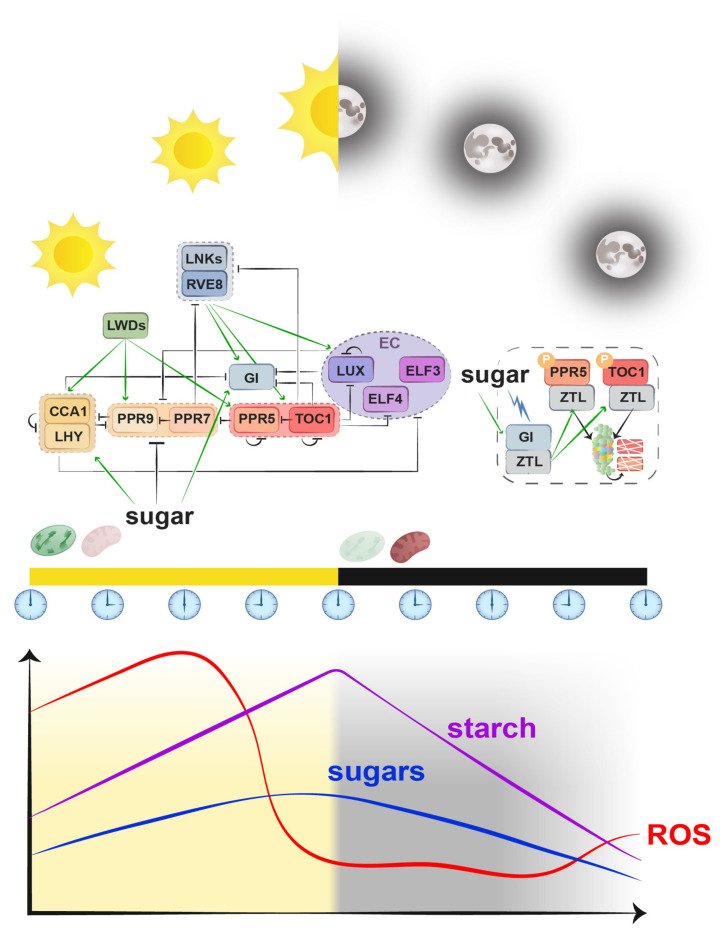
Schematic representation of transcriptomic and post-translational regulation within the plant circadian oscillator. Clock components are shown with their sequential expression across a day/night cycle (white: day; black: night) and are positioned according to peak expression time (clock faces indicate hours after dawn). Green arrows indicate transcriptional activation; black bars indicate repression. Colored dashed boxes group components by functional category. Sugars act as metabolic cues modulating specific oscillator nodes (e.g., repression of PRR7, activation of GI, and activation of CCA1/LHY). The black dashed box highlights the sugar- (green arrow) and blue light- (blue lightning) enhanced stabilization of GI, which promotes ZTL accumulation (green arrows) and subsequent phosphorylation-dependent degradation of PRR5 and TOC1 via the 26S proteasome (shredded red and orange boxes). Chloroplast and mitochondrial activity are schematized above the timeline, with faint coloring indicating reduced activity. Below, the approximate circadian accumulation patterns of starch, sugars, and ROS are shown as waveforms based on data from Annunziata et al. [21] and Lai et al. [22]. EC: Evening Complex; P: Phosphorylation.

**Figure 2 plants-14-02464-f002:**
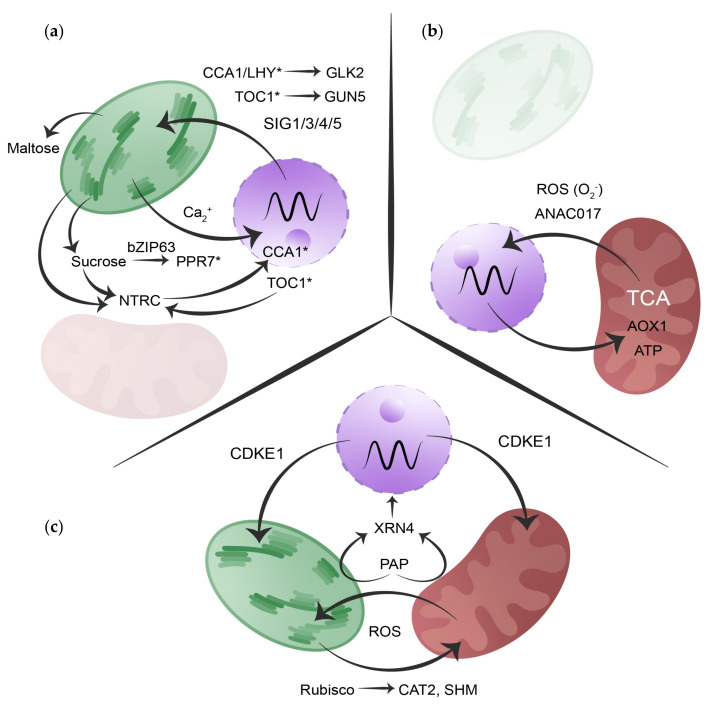
Schematic representation of the bidirectional communication between the circadian clock and organellar metabolism. A simplified depiction of the chloroplast (green), mitochondrion (red), and nucleus (purple) highlights the most representative metabolic regulatory pathways linking organellar metabolism with the circadian system. (**a**) Chloroplast-derived signals, including sugars and redox cues (e.g., NTRC), entrain the clock and are reciprocally regulated by core components (e.g., CCA1, TOC1) via GLK2, GUN5, SIG factors, and NTRC. (**b**) Mitochondrial ROS act as retrograde signals through ANAC017 to modulate clock genes, while the clock controls mitochondrial functions such as AOX1a expression, ATP synthesis, and TCA cycle activity. (**c**) Convergent retrograde signaling via PAP and ROS regulates the nuclear clock by inhibiting XRN4-mediated mRNA degradation. The photorespiratory pathway (Rubisco, CAT2, SHM) exemplifies inter-organellar coordination under circadian control. CDKE1 integrates metabolic and stress signals with clock regulation, affecting both organelles. The spiral waveform indicates the central oscillator; arrows represent regulatory direction. Core clock components are marked with an asterisk (*).

**Figure 3 plants-14-02464-f003:**
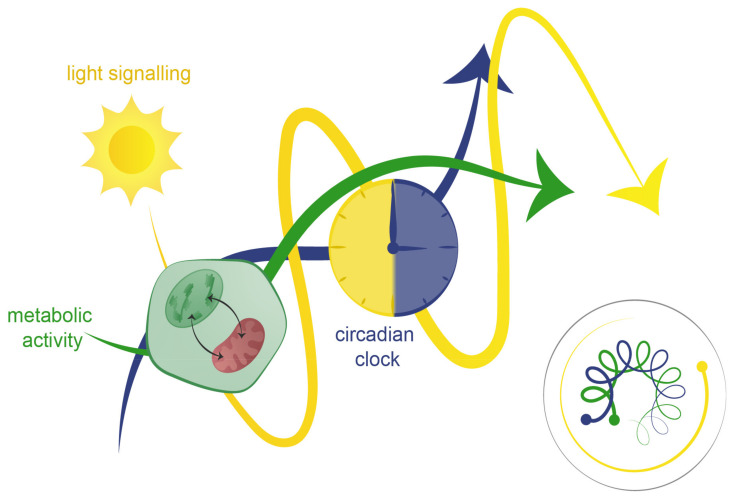
Conceptual model illustrating the dynamic equilibrium between light signaling, metabolism, and the circadian clock. The figure represents the reciprocal regulation among light signaling (yellow path), metabolic activity (green path), and the circadian clock (blue path and clock face). These three systems operate in a balanced, co-regulated loop where perturbation of one component (e.g., metabolic imbalance, altered light cues, or clock disruption) can destabilize the entire network. The small inset spiral (bottom right) visually evokes the “three-body problem”, illustrating how non-linear interactions among three mutually interacting systems can lead to complex and stable trajectories. This model highlights the necessity of synchrony between environmental signals, internal timing, and metabolic status for maintaining physiological homeostasis.

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
