# Peer review of "Metabolism in Sync: The Circadian Clock, a Central Hub for Light-Driven Chloroplastic and Mitochondrial Entrainment"

_plants, 2025, doi:10.3390/plants14162464_

Round 1
Reviewer 1 Report
Comments and Suggestions for Authors
Authors summarize a wealth of knowledge about the bidirectional control of the circadian clock and organelles, and serves as a good guideline. However, since "Metabolism" is a broad term, the title may need to be reconsidered. Please consider including organelle or chloroplast, mitochondria in the title to match the content of the manuscript. Additionally I have some concerns.
Major points
Figure 1: The black dashed box highlights sugar- and blue-light–enhanced stabilization of GIGANTEA
This box diagram is difficult to understand. Please indicate the blue light stimulation with text and clearly illustrate that blue light and sugar indirectly promote the degradation of PRR5 and TOC1 (negative regulation).
L443-451
The following paper should be cited. This demonstrated the importance of daily magnesium fluxes for timekeeping processes in a cell.
https://doi.org/10.1038/nature17407
Related to this, although no relationship with organelles has been demonstrated, the reporter analysis using duckweed has provided interesting insights into metabolic oscillations within cells. Please consider citing this.
https://doi.org/10.1093/plphys/kiad303
https://doi.org/10.1093/plphys/kiad218
Minor points
The “diel” is recommended instead of “diurnal”. Diurnal originally means active in the day.
L413 It may be better that Boix et al (2024) change to Boix et al. [143] as the same format of Haydon et al. [55] at L400.
Author Response
Comment 1: Authors summarize a wealth of knowledge about the bidirectional control of the circadian clock and organelles, and serves as a good guideline. However, since "Metabolism" is a broad term, the title may need to be reconsidered. Please consider including organelle or chloroplast, mitochondria in the title to match the content of the manuscript
Response 1: We thank Reviewer 1 for the thorough reading and valuable suggestions. We agree that including organelles in the title improves precision. We have revised the title to:
Metabolism in Sync: The Circadian Clock, a Central Hub for Light-Driven Chloroplastic and Mitochondrial Entrainment
Comment 2: Figure 1: The black dashed box highlights sugar- and blue-light–enhanced stabilization of GIGANTEA. This box diagram is difficult to understand. Please indicate the blue light stimulation with text and clearly illustrate that blue light and sugar indirectly promote the degradation of PRR5 and TOC1 (negative regulation).
Response 2: We have revised Figure 1 to explicitly indicate blue light stimulation (blue lightning symbol) in the text box and added a green arrow from sugar to GI to clarify metabolic stabilization. Additionally, we included green arrows showing ZTL-mediated PRR5 and TOC1 degradation and visually highlighted the proteasome-dependent degradation step using shredded colored boxes. These changes improve clarity and address the reviewer’s concerns.
Comment 3: L443-451. The following paper should be cited. This demonstrated the importance of daily magnesium fluxes for timekeeping processes in a cell.
https://doi.org/10.1038/nature17407
Response 3: We have included the suggested citation highlighting the importance of daily magnesium fluxes for cellular timekeeping (Feeney et al., 2016, Nature, doi:10.1038/nature17407) in the paragraph spanning lines 445–451, as recommended.
We thank the reviewer for suggesting the additional references related to duckweed. While these studies provide valuable insights into cell-autonomous and non–cell-autonomous circadian regulation, they focus primarily on tissue-level synchronization rather than organelle-derived signaling. To maintain the specific focus of our review on chloroplast and mitochondrial metabolic entrainment, we have chosen not to include these references.
Comment 4: The “diel” is recommended instead of “diurnal”. Diurnal originally means active in the day.
Response 4: We have replaced “diurnal” with “diel” in lines 343, 355, 374, and 383, as suggested, to reflect the more accurate terminology.
Comment 5: L413 It may be better that Boix et al (2024) change to Boix et al. [143] as the same format of Haydon et al. [55] at L400.
Response 5: We have corrected the citation format of Boix et al. (2024) to “Boix et al. [143]” in line 415 to maintain consistency with the format used for Haydon et al. [55].
Reviewer 2 Report
Comments and Suggestions for Authors
This review paper on photoperiod is relatively comprehensive, but it has the following problems.
- The subheadings are not clear enough, especially in 2, there is only 2.1 and no 2.2 or 2.3.
- There is a lot of content in each part. It would be best to add a subheading layer, which will make it look more explicit.
- There are too many annotations under Figures 1 and 2. They should be simplified.
- The full text has been refined and revised.
Author Response
Comment 1: The subheadings are not clear enough, especially in 2, there is only 2.1 and no 2.2 or 2.3.
Response 1: We thank the reviewer for this suggestion. We have revised Section 2 to improve clarity by adding a new subheading 2.2 (". Clock modulation of light signaling and sensitivity") in addition to the existing 2.1, thus clarifying the internal structure and enhancing readability.
Comment 2: There is a lot of content in each part. It would be best to add a subheading layer, which will make it look more explicit.
Response 2: We have carefully reviewed the manuscript and added additional subheadings where appropriate, particularly in Section 2, to improve structure and make the content more explicit and accessible to the reader.
Comment 3: There are too many annotations under Figures 1 and 2. They should be simplified.
Response 3: We thank the reviewer for this suggestion. While we agree that figure annotations should be concise, other reviewers explicitly requested more detailed explanations for Figure 1 to improve clarity. To balance these perspectives, we decided to maintain detailed annotations in Figure 1 to guide the reader through its complex schematic interactions, while summarizing and simplifying the caption for Figure 2 as suggested. We believe this level of detail is important to accurately convey the integrated regulatory interactions discussed in the manuscript.
Comment 4: The full text has been refined and revised.
Response 4: We thank the reviewer for their positive and supportive comments.
Reviewer 3 Report
Comments and Suggestions for Authors
Luis Cervela-Cardona and colleagues present in this paper the review of circadian clock as a central hub for light driven metabolic entrainment, which integrates light and metabolic signals to optimize growth and stress responses.
The manuscript is well-organized with clear logic, the data and references are comprehensive and well-integrated. Considering all these aspects, I believe the paper is worthy of publication in its present form.
Author Response
Comment 1: Luis Cervela-Cardona and colleagues present in this paper the review of circadian clock as a central hub for light driven metabolic entrainment, which integrates light and metabolic signals to optimize growth and stress responses.
The manuscript is well-organized with clear logic, the data and references are comprehensive and well-integrated. Considering all these aspects, I believe the paper is worthy of publication in its present form.
Response 1: We thank Reviewer 3 for their positive and encouraging comments. We are pleased that they found the manuscript well-organized and suitable for publication.
Reviewer 4 Report
Comments and Suggestions for Authors
The review paper by Cervela-Cardona et al. entitled "Metabolism in Sync: Circadian Clock as a central hub for Light 2 driven Metabolic Entrainment" is very well written, in good English and reads nicely. The main theme of the review is to provide insight into the reciprocal links between light, plant metabolism and cirardian clock regulation under the hypothesis that the metabolic signals from chloroplasts and mitochondria are important entrainers of the circardian clock machinery. The structural organization of the paper is logical and divided into 3 levels, first describing the basic light and clock relationship, second - the main part - describing the state of knowledge for clock-driven regulation of the organelles and reciprocal influence of the organelle-generated metabolites on circardian clock and third - briefly addressing the overall interactions between light, clock and metabolism, highlighting the complexity of the system and the challanges related to its understanding. The third part (also Figure 3) is a bit general and would probably benefit from some more insightful discussing the perspectives and directions for future research (the last akapit of Section 4). For example, I missed at least mentioning about the emerging role of long noncoding RNAs in both circardian clock regulation (well described in mammals but also found in plants) and metabolism regulation. Wouldn't it be a good direction to explore this path in context of the interplay between the clock an metabolite signallig?
Regardless of this remark, in my opinion the submitted manuscript will provide a valuable contribution to the field as a comprehensive and hypothesis-driven review and I recommend its acceptance.
I also noticed minor errors:
L340: is: encodes; should be: encode
L401: is: could shortened; should be: shortened or could shorten
Author Response
Comment 1: The review paper by Cervela-Cardona et al. entitled "Metabolism in Sync: Circadian Clock as a central hub for Light 2 driven Metabolic Entrainment" is very well written, in good English and reads nicely. The main theme of the review is to provide insight into the reciprocal links between light, plant metabolism and cirardian clock regulation under the hypothesis that the metabolic signals from chloroplasts and mitochondria are important entrainers of the circardian clock machinery. The structural organization of the paper is logical and divided into 3 levels, first describing the basic light and clock relationship, second - the main part - describing the state of knowledge for clock-driven regulation of the organelles and reciprocal influence of the organelle-generated metabolites on circardian clock and third - briefly addressing the overall interactions between light, clock and metabolism, highlighting the complexity of the system and the challanges related to its understanding. The third part (also Figure 3) is a bit general and would probably benefit from some more insightful discussing the perspectives and directions for future research (the last akapit of Section 4). For example, I missed at least mentioning about the emerging role of long noncoding RNAs in both circardian clock regulation (well described in mammals but also found in plants) and metabolism regulation. Wouldn't it be a good direction to explore this path in context of the interplay between the clock an metabolite signallig?
Regardless of this remark, in my opinion the submitted manuscript will provide a valuable contribution to the field as a comprehensive and hypothesis-driven review and I recommend its acceptance.
Response 1: We thank Reviewer 4 for this thoughtful suggestion regarding long noncoding RNAs. We agree that lncRNAs represent an interesting and emerging area in circadian and metabolic regulation. However, to the best of our knowledge, current evidence specifically linking lncRNAs to chloroplast and mitochondrial signaling in plants is still limited. The interplay between lncRNAs, metabolism, and the circadian clock is indeed an exciting field that well deserves a dedicated review of its own. Therefore, we have decided not to include this topic in order to maintain the focused scope of our review. We believe this ensures a clear and coherent narrative centered on organelle-derived metabolic entrainment.
We also greatly appreciate the reviewer’s positive assessment of our manuscript as a valuable and hypothesis-driven contribution to the field.
Comment 2:
Response 2: We thank the reviewer for carefully noting these minor errors. We have corrected “encodes” to “encode” in line 339 and revised “could shortened” to “could shorten” in line 400.
Round 2
Reviewer 2 Report
Comments and Suggestions for Authors
All the questions I raised have been revised.